

# Who is a Migrant? Abandoning the Nation-state Point of View in the Study of Migration

Stephan Scheel[1*] and Martina Tazzioli[2]

**1** University of Duisburg-Essen
**2** Goldsmiths, University of London
* Corresponding Author: stephan.scheel@uni-due.de

Saturday, March 5, 22

## Abstract

**This article develops an alternative definition of a migrant that embraces the perspective of mobility. Starting from the observation that the term 'migrant' has become a stigmatizing label that problematizes the mobility or the residency of people designated as such, we investigate the implications of nation-state centered conceptions of migration which define migration as movement from nation-state A to nation-state B. By asking 'Who is a migrant in Europe today?' we show that nation-state centered understandings of migration rest on a deeply entrenched methodological nationalism and implicate three epistemological traps that continue to shape much of research on migration: first, the naturalization of the international nation-state order that results, secondly, in the ontologisation of 'migrants' as ready-available objects of research, while facilitating, thirdly, the framing of migration as problem of government. To overcome these epistemological traps, we develop an alternative conception of migration that, inspired by the autonomy of migration approach, adopts the perspective of mobility while highlighting the constitutive role that nation-states' bordering practices play in the enactment of some people as migrants. Importantly, this definition allows to turn the study of instances of migrantisation into an analytical lens for investigating transformations in border and citizenship regimes.**

## 1. Introduction

In August 2015 *Al Jazeera* announced it would no longer use the term 'migrant' to designate people trying to cross the Mediterranean in overcrowded boats, calling them 'refugees' instead. The news agency explains this move on its webpage as follows: 'The umbrella term migrant is no longer fit for purpose when it comes to describing the horror unfolding in the Mediterranean. It has evolved from its dictionary definitions into a tool that dehumanizes and distances, a blunt pejorative' (Malone, 2015). Following this view, the word *migrant* has become a toxic term that should be abandoned because it stigmatizes people labelled as such.

Similar observations have been made by critical migration studies scholars. Bridget Anderson notes that, the term *migrant* is not reducible to a neutral description of persons crossing international borders: '"migration", she argues, signifies problematic mobility.' Accordingly, 'not all mobility is subject to scrutiny, but "migration" already signals the need for control and in public discourse is often raced and classed' (Anderson, 2017: 1532). Since migration, and in particular the mobility of the poor, is

regulated through laws on citizenship and notions of national belonging, the historically and geographically contingent problematization of the mobility and presence of some people as 'migration' can be used as an analytical lens to study transformations in migration politics and related border and citizenship regimes. Hence, Anderson (2017: 1535) calls for turning 'the problematization of migration into a tool for inquiry.'

In this article we follow Anderson's call of 'problematizing the problem of migration' (ibid.) by starting from the apparently banal question: Who is a migrant in Europe today? We engage with this question to expose and challenge 'the nation-state point of view of spatial mobility' (Favell, 2007: 271) which underpins the framing of migration as a problem requiring constant monitoring as well as governmental interventions of regulation and control. This nation-state point of view is carried by dominant understandings of migration as movement to and residence in nation-state B from nation-state A. The latter informs policy-making as well as statistical and academic knowledge production on migration. The United Nations (UN) define a migrant, for instance, 'as a person who moves to a country other than his or her usual residence for a period of at least a year' (UN, 2002: 11).[1] Nation-state centered understandings of migration also dominate the thinking of wider publics about migration, thus shaping migration-related political debates. The entry in the *Miriam-Webster Online Dictionary* stresses, for example, that 'to immigrate' would mean 'especially: to come into a country of which one is not a native for permanent residence.'[2] Such nation-state centered understandings of migration rest on a deeply entrenched methodological nationalism that implicates three epistemological traps which continue to shape much of contemporary research (and political debate) about migration: first, the ontologization of 'migrants' as ready-available objects of research, which goes hand in hand with, secondly the naturalization of the 'national order of things' (Malkki, 1995) that facilitates, thirdly, the framing of migration as problem of government in need of close monitoring and interventions of regulation and control.[3]

In brief, methodological nationalism implies a conception of societies as nationally bounded containers (Wimmer & Glick-Schiller, 2003). Within this 'container thinking' migrants can only emerge as intruders who disturb and endanger the alleged cultural homogeneity and social equilibrium of the imagined community of national citizens. What slips into the background are the many practices of bordering and boundary-making through which some people are enacted, problematized and targeted as migrants.[4] By placing these processes of migrantisation at the centre of attention, we pursue two interrelated objectives with this article.

First, we want to overcome the three epistemological traps implicated by nation-state centred conceptions of migration. To this end, we develop an alternative conception of migration that highlights the 'making of migration' (Tazzioli, 2020), that is, the practices of bordering and processes of boundary-making through which some people are enacted as migrants.[5] We propose an alternative definition of a migrant that is inspired by the autonomy of migration (AoM) literature (Mezzadra, 2011; Papadopoulos, Stephenson, & Tsianos, 2008; Scheel, 2019): we understand a *migrant as a person who, in order to move to or stay in a desired place, has to struggle against bordering practices and processes of boundary-making that are implicated by the national order of things*. This definition adopts the perspective of mobility and puts 'border struggles' (Mezzadra & Neilson, 2013) at the centre of the analysis. Importantly, by proposing to adopt the perspective of mobility in the definition of a migrant we neither want to erase the multiplicity of migrant conditions, nor do want to suggest that there is a single migrant perspective. To the contrary, if we start the analysis with border struggles and ask who is enacted as a migrant in this particular situation, migration emerges as something that is contingent, relational and multiple. Thus, the focus on migrants' border struggles as a key element of our definition highlights that there are 'a myriad of ways to be "migrants" (Mezzadra, 2011) which are shaped by lines of age, class, gender, 'race'", sexual orientation and so forth (Scheel, 2019) and that there exists, consequently, only a plurality of migrant perspectives. In this way, and this is our second objective, our definition challenges the essentialisation and de-historization of 'migrants' as a stable sociological category. Adopting the perspective of mobility in the study of migration thus fractures the category of the migrant while also putting it on the move.

It should be noted that this intervention is not only directed at mainstream migration studies, that is, scholarship that uses nation-state centric understandings of migration, such as those carried by the UN-definition cited above, as a starting point of research. We also want to contribute to lines of

thought and inquiry that are critical and reflexive in regards to their object of study. Since the publication of Andreas Wimmer's and Nina Glick-Schiller's (2002, 2003) seminal work on methodological nationalism, various strands of scholarship have developed inspiring suggestions of how to overcome this epistemic bias in the study of migration. Examples include the proposal of a transnational paradigm that moves beyond the national-container model of society by studying transnational networks, connections and social spaces of 'in-betweenness' that migrants forge by living 'here' and 'there' (Glick-Schiller, Basch, & Szanton Blanc, 1995). Yet, as we elaborate below, the transnational paradigm remains haunted by methodological nationalism, because the (criss-)crossing of national dividing lines still remains the defining feature of who a migrant is. Also the proposal of scholars working with the 'new mobilities paradigm' (Büscher & Urry, 2009) fails to offer a viable solution. Their proposal to de-exceptionalize migration by understanding it as one form of mobility among many others essentially suggests to ignore the continued relevance of practices of bordering and boundary-making implicated by the national order to things. The latter cannot simply be ignored because they do have very real consequences for people that are labelled and targeted as migrants. What is needed to resolve this conundrum is an alternative conception of a migrant that starts from the perspective of mobility in order to transcend the epistemic traps implicated by state-centric definitions of migration.

In this article we develop such an alternative definition in three moves. In the first section we show that the nation-state point of view began to dominate understandings of migration from the 1920s onwards before explaining how methodological nationalism and the epistemological traps implicated by it continue to shape much of the research on migration. Based on a review of the most important existing attempts to overcome methodological nationalism and statist conceptions of migration, the second section develops an alternative definition of migration from the perspective of mobility. The third section illustrates through three empirical examples how our AoM-inspired definition of a migrant can be put to use in order to demonstrate its analytical and political surplus value.

The three examples we chose relate to processes of migrantisation implicated by (1) the Schengen visa regime of the European Union (EU), (2) the integration paradigm and (3) the bordering of Europe's southern frontier in the Mediterranean. We chose these three cases to highlight the wide variety of practices of migrantisation and related processes of bordering and boundary drawing which cannot be captured by state-centric understandings of migration. Moreover, each case allows us to highlight particular aspects of our alternative definition of a migrant: the Schengen visa regime illustrates that processes of migrantisation operate along lines of class, race, age and gender and that people try to escape their migrantisation in multifarious ways. The integration paradigm highlights, in turn, that migrantisation is strongly intertwined with processes of racialization and that migrantisation is, consequently, a matter of degrees. The study of processes of migrantisation operating in Europe's southern borderzone shows, in turn, that the enactment of some people as migrants is both relational and contingent.

Finally, our three cases studies highlight that we developed our definition of a migrant in the context of our research on Europe and its border-zones. Hence, it is key to clarify that our alternative conception of a migrant is a situated one. And while the main contribution of this paper is theoretical, we do not intend to provide a universal, definite answer to the question "who is a migrant?". We rather hope that our alternative definition will be put to work and nuanced in light of the specific migration contexts. At the same time we are confident that our definition and its impetus to re-focus the analysis on processes of migrantisation provide useful epistemic-methodological points for the study of migration, border and citizenship politics beyond Europe, not the least because – contrary to the dominant narrative – Extra-European states were, historically, the first to deploy territorial immigration controls and to embrace – in the context of processes of decolonization – the nation-state point of view on migration (Vigneswaran, 2020).

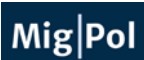

## 2.   Methodological Nationalism and the Study of Migration

With his *Laws of Migration* E.G. Ravenstein (1885) provided the first attempt of a systematic analysis and theorization of migratory movements. Ravenstein's analysis focused on migration in the United Kingdom and was based on the results of the 1881 census. What is striking about Ravenstein's analysis is that it did not distinguish between internal and international migration. Instead, Ravenstein treated all population movements – no matter if they involved the crossing of national borders between one of the three kingdoms constituting the UK at that time or only the crossing of administrative boundaries between counties – as part of the same phenomenon (Wimmer & Glick-Schiller, 2003: 587). What this example highlights is that the predominance of the nation-state point of view in the study of migration and the conflation of migration with 'international migration' is a relatively recent development.

To this regard, Yann Stricker (2019, p. 469) shows how the consolidation of the category 'international migration' in population statistics was interrelated 'with a shift from an imperial to an internationalist point of view' on human mobility. Attempts of the International Labor Organization (ILO) to produce statistics on people on the move – most notably workers – on a global scale raised concerns about the meaning of borders within the British Empire, which comprised colonies, protectorates, dominions, mandates and the British Raj at that time. The ILO's request to provide data on the movements of workers who cross an *international border* was greeted with great skepticism by British officials who insisted that movements within the empire were not international in character. The underlying fear was that the use of national dividing lines in the conception of migration by the ILO and the production of respective migration statistics could fuel claims for independence of nationalistic movements within the British and other colonial empires (Stricker, 2019: 475-476). Hence, British officials insisted on labelling emigration from the United Kingdom to the dominions and colonies – which was promoted by the UK government after the First World War – as 'oversea settlement'.  This reluctance of officials to consider mobility within the British Empire as 'migration' is today echoed by the insistence of the European Commisison to consider mobility between Schengen member states not as 'migration' but as the mobility of EU citizens enjoying their 'freedom of movement.' By showing that the emergence of the nation-state point of view on migration is a relatively recent development Stricker's careful analysis demonstrates that conceptions of migration are contingent and thus contestable and contested.

Notwithstanding efforts of British officials to safeguard the imperial view on human mobility, the nation-state point of view became hegemonic after the First World War. Processes of nation-building fostered a new conception of 'the people' along ethnic and/or racial lines which began to replace a "civic" notion of peoplehood. '"The people" began to mean a nation united by common ancestry and a shared homeland, no matter where its members might have wandered' (Wimmer & Glick-Schiller, 2003: 587). As a consequence, migrants began to be imagined as essentially different subjects who continued to hold memberships of their ancestral homelands. In brief, migrants began to be conceived as 'foreigners'. Hence, the consolidation of the 'national order of things' (Malkki, 1995) heralded the emergence of nation-state point of view as the dominant perspective on migration.  By the end of the First World War, migrants were 'seen as politically dangerous and nationally or racially fundamentally different others' whose presence endangered the isomorphism between the imagined community of (national) citizens, the sovereign state order and its territory (Wimmer & Glick-Schiller, 2003: 589). In the social sciences the conception of society as a social order contained within the territorial limits of the nation-state became the unquestioned, often implicit starting point of social theory and empirical research.). In other words, methodological nationalism became the modus operandi of most of the social sciences, including the study of migration.

Methodological nationalism has been identified as a complex epistemic bias that continues to shape the research agendas and conceptual frameworks of entire disciplines. Andreas Wimmer and Nina Glick-Schiller distinguish between three variants of methodological nationalism: First, a wide-spread *ignorance* of how nationalism and the formation of nation-states has been shaping some of the most important concepts of social and political theory. In brief, '[n]ation-state principles were so routinely structured into foundational assumptions of theory that they vanished from sight' (Wimmer & Glick-Schiller, 2003: 579). This is also the case for state-centric understandings of migration, as we explain

below. Second, a *naturalization* of the modern nation-state as the universal mode of political organisation and belonging by 'taking for granted nationally bounded societies as the natural unit of analysis (579). In this way 'naturalization [has] produced the container model of society that encompasses a culture, a polity, an economy and a bounded social group' (ibid). This 'container-thinking' underpins, third, the *territorial limitation* of social scientific analysis to the boundaries of the nation-state (Wimmer & Glick-Schiller, 2002: 307). Through the conception of nation-states as spatial containers of society '[t]he nation-state and modern society become conceptually as well as historically indistinguishable' (Chernilo, 2011: 99). Importantly, this 'territorial trap' (Agnew, 1994) continues to shape research questions, theories and debates of entire disciplines. To this Speranta Dumitru (2014) adds a fourth form of methodological nationalism that Roger Brubaker calls groupism: the tendency to conceive of groups, often along ethnic, national or racial lines, as 'internally homogeneous and externally bounded entities' and 'fundamental units of social analysis' (Brubaker, 2002: 164). This form of methodological nationalism has become a cornerstone of a whole branch of migration studies as it underpins the 'integration paradigm', as we explain in more detail below. In brief, groupism supports territorialised understandings of culture and (national) identity and the related conception of migrants (and their descendants) as 'people not from here' in need of integration. Due to this form methodological nationalism, migration is no longer exclusively understood as international movement of people, but increasingly also as a (inherited) feature of an individual (Renard, 2018).

Scholars import all four variants of methodological nationalism into their research if they adopt state-centred understandings of migration. By state-centred we mean conceptions of migration that make the division of the world into a set of mutually exclusive nation-states the unquestioned reference point for the determination of what migration is. Such conceptions of migration as movements from nation A to nation B result from the *ignorance* of how the formation of modern nation-states has influenced predominant understandings of migration, which in turn help to reify the *naturalisation* of the national order of things. As a result, state-centered conceptions of migration silently accept the claimed prerogative of nation-states to control access to their territories. Ultimately, statist conception of migration thus invisibilize nation-state practices of bordering and boundary-making that enact some people as migrants in the first place.

The consolidation of the nation-state point of view on migration also nurtures the still dominant idea that migrants are 'uprooted' and need to be integrated in the nationally bounded receiving society which is viewed as culturally homogeneous (Wimmer & Glick-Schiller, 2003: 591). Hence, state-centred conceptions of migration continue to carry the idea of a stark distinction between (native) citizens and (foreign) migrants in need of 'integration'. This is also why the continued use of hegemonic state-centred understandings of migration is problematic for critical migration studies scholars: they import this idea, even if only implicitly, back into their research.

The tacit assumptions carried by state-centered conceptions of migration have significant consequences as they implicate three epistemological traps that continue to shape much of the research on migration. First, the adoption of state-centred understandings of migration implicates an ontologization of migrants as ready-available subjects of research. This ontologization of migrants is coupled with the reification of a binary distinction between (foreign) migrants and (native) citizens along lines of (citizenship) status and (national) belonging. It often results in a kind of 'migrantology' (Römhild, 2017: 70) which reduces migration research to the study of migrants, their practices, cultural preferences, experiences and so forth, which are assumed to be distinct from the outset. Secondly, state-centred conceptions of migration naturalise the national order of things, which in turn invisibilises the discourses of belonging, practices of bordering, legal norms and so forth that enact some people as migrants, as we have explained above. By representing nation-states as passive spatial units that are crisscrossed by migratory movements state-centred definitions of migration 'obscures that the modern state and system of states have helped [and still help] to produce what they seek to contain: international migration' (Joppke, 1998: 5). Thirdly, the adoption of state-centred understandings of migration reifies the framing of migration as a security issue in need of close monitoring, regulation and control. Due to the conception of societies as nationally bounded containers, migrants emerge as disruptive factors i.e. as 'intruders' who disturb and potentially destroy the imagined isomorphism between the people and the nation which is, at once, understood as a culturally homogenous community, a group of solidarity and a citizenry that votes and is represented by the sovereign (Wimmer & Glick-Schiller, 2003). In this way, the container-thinking underpinning state-

centred conceptions of migration contributes to the securitization of migration. Didier Bigo underlines this effect of methodological nationalism*,* arguing that 'the securitization of the immigrant as a risk is based on our conception of the state as a body or a container of the polity' (2002: 65).

## 3.    Who is migrant? De-naturalizing the national order of things

Since the pioneering work of Wimmer and Glick-Schiller (2002) numerous scholars have made proposals on how to transcend methodological nationalism in the study of migration. In this context methodological transnationalism, i.e. the study of practices, connections and communities that crisscross international borders, is one of the most influential approaches (Amelina & Faist, 2012). A transnational methodology 'tries to capture how they [migrants] simultaneously become part of the places where they settle and stay connected to a range of other places at the same time' (Levitt, 2012: 495). In this way transnationalism permits to move beyond the national container model of society and the territorial limitation implicated by it. However, scholars of transnationalism often craft their unit of study as a bounded 'migrant' community that is defined by a shared identity along lines of ethnicity and nationality (Glick-Schiller, 2010: 111).

This form of 'groupism' (Brubaker, 2002) is particularly pronounced in diaspora studies. It is basically the epistemic starting point and modus operandi of a field of study investigating the identities, experiences and practices of distinct groups of people defined along ethnic or national lines that have been dispersed across several geographic locations and even continents through expulsion, colonial conquest and slavery or armed conflict. The treatment of these people as a distinct group of 'diasporic people' is justified with the assumption of a shared 'long distance nationalism' (Benedict Anderson, 2006) which is often based on a shared experience of eviction and displacement and supposedly functions as the pre-dominant source of identity for a *diaspora* of people.  It motivates them to engage in cultural and social activities as well as political mobilizations whose central reference point is a 'lost home' or ancestral territory (Banerjee, MacGuisness, & McKay, 2012). Ultimately, diaspora studies overcome one form of methodological nationalism – territorial limitation – by embracing another one –groupism. They do so by adopting a transnational analytical framework that follows cross-border connections, networks, social practices and political mobilizations of one particular group defined along ethnic or national lines. Moreover, many studies continue to use nation-states as units of analysis by analyzing and comparing the practices of one diaspora in two or more host-states, as Maria Koinova (2021) succinctly observes in her comprehensive overview of field.

However, even studies of transnationalism that succeed in avoiding groupism remain haunted by methodological nationalism. The reason is that an expansion of the scope of the analysis beyond the national container does little to move scholars beyond statist understandings of migration (Favell, 2007: 270). For '[g]oing beyond methodological nationalism in the study of current migration thus may require more than a focus on transnational communities instead of the nation and its immigrants' (Wimmer & Glick-Schiller, 2002: 324). The same can be said of approaches that try to transcend methodological nationalism by simply shifting the analytical focus from the national to the local (Glick-Schiller & Çağlar, 2009) or global scale (Glick-Schiller, 2010), because the (criss-)crossing of national borders still remains the defining criterion for determining who a migrant is

The next approach that we discuss goes beyond a simple alteration of the spatial focus of the analysis. It aims at the 'de-migranticisation of research on migration and integration' (Bojadžijev & Römhild, 2014; Dahinden, 2016). Following Janine Dahinden (2016: 2209), migration and integration research constantly confirm 'the idea of migrants as different from citizens and the perceived need for nation states to manage this difference [...]' (2209). The reproduction of migration as a category of difference happens in particular when scholars use 'migration or ethnicity as *the* central criterion of difference in research questions, research design, data collection, analysis and theory [...]' (2211). Hence, Dahinden proposes three strategies to de-migranticize research on migration: First, she proposes to clearly distinguish between common-sense categories of migration as they are used by actors in the everyday, particularly in migration policy discourse, on the one hand, and analytical research categories on the other hand (2213). Second, Dahinden suggests to align migration theory more closely with other social science theories as a way to de-exceptionalise migration (2214). Third

she calls for re-orienting the focus of analysis away from 'migrant populations'. Rather than distinguishing between 'migrant' and 'non-migrant' populations from the outset, Dahinden (2016, p. 2218) suggests to begin the analysis with 'overall populations' and to make it a question of empirical inquiry if and how 'migration and ethnicity [matter] in the phenomenon being investigated.'

While we share Dahinden's assessment of migration studies, we have doubts about her proposal to de-migranticize research on migration. The reason is that, as Dahinden acknowledges herself, differences between migrants and non-migrants do exist as 'empirical facts' precisely because of the existence of a state migration apparatus that 'creates specific social realities and inequalities' (2016: 2211). Rather than simply bracketing these differences through an analysis that 'investigate[s] social processes in general and then evaluate[s] the role of migration and ethnicity in them' (Dahinden, 2016: 2213), we propose a framework that studies how, and through what kind of practices, some people are constituted and governed as migrants. What is needed is not a de-migrantization of migration research, but a conception of migration that accounts for the *making of migration* (Tazzioli, 2020) - that is for the political and legal processes of *migrantization* that are inherent to the national order of things.

With 'migrantisation' we refer to the enactment of certain subjects as 'migrants', that is, as 'people out of place' who do not (really) belong to the places and societies they inhabit (Sharma, 2020, p. 4). Processes of migrantisation involve manifold practices of bordering and boundary-making that nation-states rely on in order to establish and reproduce themselves as a bounded territory, people and jurisdiction. Importantly, processes of migrantisation are heavily intertwined with processes of racialization without being reducible to the latter. The reason is that the figure of the migrant has become a substitute for the biological notion of race in racist discourses and practices with the onset of the era of decolonization and the reversal of population movements between the (former) colonies and the (former) colonial powers (Balibar, 1995a). In the post-colonial world divided into a set of mutually exclusive nation-states, migrants are constituted as the 'quintessential Other' and 'made to be outside of the nation even as they live on national territory' (Sharma, 2020: 4). This enactment of migrants 'as the others of National-Natives' (13) often features processes of racialization (Balibar, 1995b). Hence, a focus on processes of migrantisation means putting at the core of the analysis the racializing mechanisms through which some people are turned into 'migrants' and the colonial legacies of the racialised governing of mobility.

Indeed, the racialized category of the migrant has historically been used to designate people from former colonies. The migrantisation of subjects from the British Empire was, for example, 'prefigured by imperial needs to discipline and contain a labor force freed from slavery' (Sharma, 2020: 25). In the case of France, racialized notions of national belonging allowed to frame colonial subjects as 'indigenous' nationals without full citizen rights, thus prefiguring the conception of the same people as (racially and culturally different) 'foreigners' from the beginning of the period of decolonization in the 1960s onwards (Spire, 2020). What these examples illustrate is that the enactment of (some) people as migrants often features processes of racialization fraught with histories of colonization and de-colonization. Nevertheless, it is important to retain a distinction between processes of migrantisation and racialization and to consider 'national peculiarities, context-specific moments and interactions with other power relations, like classism, sexism and queer/transphobia etc.' (Tudor, 2018: 1058) in the analysis of the relationship between the two.

Before we introduce an understanding of migration that accounts for processes of migrantisation, we briefly discuss 'the new mobilities paradigm' (Büscher & Urry, 2009) as another important approach seeking to challenge methodological nationalism. On the back of often enthusiastic accounts of globalization, scholars like John Urry (2007) or Tim Cresswell (2006) claim that the world has become more mobile. Accordingly, the social sciences need a new conceptual and methodological apparatus that allows to focus not on stability and stasis as the normal state of affairs, but on mobile flows and cross-border connections. This 'mobility turn' requires to abandon the national container model of society (Urry, 2001) and to conceive of migration as one form of travel and movement among many (Urry, 2007: 10-11). However, the subsumption of all kinds of movement and travel under one single analytical category (mobility) ignores how the continued relevance of national borders and ethnic boundaries still shapes people's highly differentiated access to and experience of mobility (Glick-Schiller & Salazar, 2013; Kalir, 2013; Samers, 2010; Wimmer & Glick-Schiller, 2002).

In this context it is important to mention Thomas Nail's book *The Figure of the Migrant*, which is also situated in the mobility turn literature. The book develops a reconceptualization of the figure of the migrant from the viewpoint of movement. What migrants share, according to Nail, is that 'their movement results into a certain degree of expulsion from their territorial, juridical or economic status' (2015: 72). By defining migration in relation to both movement *and* expulsion, Nail moves beyond a linear account of mobility. Yet, by positing the migrant as the political subjectivity of contemporary societies, Nail tends to reify the 'migrant' as the paradigmatic figure of the present. As a result, migration is diluted to a generalized increased mobility, detached from the materiality of migrants' struggles for moving and staying in their desired place as well as from the racialized mechanisms of discrimination upon which the *making of migration* is predicated.

What all these approaches and proposals share is the attempt to decenter the focus on migration through a broader analysis of mobilities more generally. In our view, this move does however little to transcend statist conceptions of migration. Actually, a similar concern has also been raised within the field of migration studies, where scholars have criticized state-centered categorizations of migration, challenging in particular its spatial and temporal criteria and related distinctions between internal and international or temporal and permanent migration (Collyer & Haas, 2012). Others have criticized related politics of labelling and classification and their consequences for the labelled (Crawley & Skleparis, 2018; Zetter, 2007). Yet, the search for less discriminatory labels or for alternative definitions does not necessarily unsettle the nation-state point of view on migration, nor the taken-for-granted idea that migration should be defined in governmental terms – that is, as a phenomenon to be governed.

What distinguishes migration from other forms of mobility is that it is the fabrication of clashes with practices of statecraft. 'It is precisely the control which states exercise over borders that defines international migration as a distinct social process' (Zolberg, 1989: 405). Nation-states do not just shape migration via their policies. They constitute it. This is why Abdelmalek Sayad (2004) aptly describes the modern nation-state as a vast discrimination machine that, in order to reproduce itself, draws and polices a clear demarcation line between those who belong to the national citizenry and those who do not. These consist in the manifold practices, devices, actors, institutions, discourses, sites, technologies of bordering that are mobilized to draw this distinction and which enact migration as a intelligible reality. Nicholas De Genova aptly summarizes this observation as follows: 'it is the bordered definition of state territoriality that constitutes particular forms *and* expressions of human mobility as "migration" and classifies specific kinds of people who move as "migrants". *Borders make migrants'* (De Genova, 2015: 4; italics in original). Without borders, there would be neither migration nor migrants, but only mobility and people on the move (De Genova, 2013: 253). It is this intimate and mutually constitutive relationship between migration and the bordering practices of nation-states which distinguishes migration from other forms of mobility.

Hence, we need to bring attention to what is invisibilised by state-centered conceptions of migration: the practices of bordering through which nation-states constitute and govern some people as migrants in order to reproduce themselves as territorially-bounded, culturally distinct, imagined communities and sovereign orders. We therefore propose to invert, as suggested by authors like Kalir (2013) or Bassaram and Guild (2017), the nation-state centered perspective of statist conceptions of migration. But in contrast to the former we place the practices of bordering through which nation-states employ to govern some people as migrants center-stage. We achieve this by taking inspiration from the AoM literature which calls on scholars to investigate contemporary border, migration and citizenship regimes from migrants' perspective (Mezzadra, 2011; Papadopoulos et al., 2008; Scheel, 2019). As suggested by its name, the AoM's central hypothesis attributes moments of autonomy, that is moments of uncontrollability and excess, to migratory practices and movements. Originally developed as a counter-narrative to the politically problematic metaphor of *Fortress Europe*, the AoM has been developed into a heuristic model that permits scholars to investigate contemporary border regimes and migratory processes from migrants' perspective with a particular focus on their 'border struggles' (Mezzadra & Neilson, 2013). However, AoM-scholars have so far not sufficiently considered the implications that the inversion of the state-centred perspective has for the conception of who a migrant is. The adoption of the perspective of mobility in the study of migration makes it, indeed, necessary to abandon the nation-state centred definition of migration as movement from one

national container to another one. Therefore, the following proposal of an alternative definition of a migrant from the perspective to mobility is also meant as a contribution to the AoM-literature.

Inspired by the AoM, our alternative definition focuses on the *struggles* people have to engage in to move to or stay in a desired place. These struggles are 'border struggles' because they 'take shape around the ever more unstable line between the "inside" and "outside", between inclusion and exclusion' (Mezzadra & Neilson, 2013: 13). In this context it is important to note that we attribute a wide meaning to the notion of 'struggle', which does not necessarily imply a literal fight. It refers primarily to the efforts that people who are addressed and targeted as (potential) migrants have to undertake to access mobility and to defend their (contested) presence as people considered 'out of place'. This implies, in turn, that not all people subjected to border controls or processes of boundary-making are migrants according to our definition. Only if people's presence in or right to move to a desired place is denied or called into question because they are considered 'as the others of National-Natives' (Sharma, 2020: 13) these people will qualify as migrants according to our definition. Hence, migrants' struggles revolve around the clandestine subversion, evasion and mitigation of border controls and processes of boundary-making as well as the appropriation of social, economic and political rights and resources. We therefore propose to understand a *migrant as a person who, in order to move to or stay in a desired place, has to struggle against bordering practices and processes of boundary-making that are implicated by the national order of things.*

It is important to emphasize that we do not intend to reduce all migrants to one singular migrant condition by proposing this definition. In fact, migrants' struggles can take on a wide variety of forms, depending on their subject position in terms of class, 'race', gender, sexuality, nationality and age and the kind of bordering practices and processes of boundary-making they encounter (Scheel, 2019). Hence, by focusing on migrants' border struggles and by inviting scholars to begin their investigation by asking who is enacted as a migrant through what kind of practices of bordering and boundary-making in the situation under study our definition thus both fractures the category of the migrant while also putting it on the move. As a result, migration becomes a reality that can only exist as something that is contingent, relational, contested and multiple. Before we provide some examples of how our definition might be put to use to demonstrate its analytical surplus value, we briefly want to explain how our definition moves beyond statist conceptions of migration and the epistemological traps implicated by them.

First, it abandons the nation-state point of view on spatial mobility carried by statist conceptions of migration though the adoption of mobile subjects' perspective. It thus permits scholars to de-naturalize the existence of nation-states by exposing their intrinsic logic to discriminate between native citizens and migrant others through practices of bordering and boundary drawing. In so doing, our definition re-directs scholarly attention from 'migrants' to the *making of migration*, that is, to processes of migrantisation that enact and govern some people as migrants in the first place. In this way, our definition moves, second, beyond the ontologization of migrants as ready-available objects of research. By exposing how people are enacted as migrants in multifarious, situated ways, our definition puts the very category of *migrant* into motion, grasping the legal, political and material struggles that shape the migrant condition in its heterogeneity and singularity. Finally, the reversal of the nation-state point of view, if combined with a focus on border struggles, also allows to transcend the third epistemological trap of methodological nationalism: the framing of migration as problem of government. Instead of seeing (and problematizing) migration like a state, to paraphrase James Scott (1999), we believe that such an understanding of migration as intertwined with practices of bordering and related border struggles enables scholars to see (and problematize) both the state and the 'national order of things' (Malkki, 1995) from the viewpoint of migrants.

## 4.   Migration from the Perspective of Mobility: Studying Processes of Migrantisation

In this section we want to show how our definition of a migrant can be used in practice to demonstrate its analytical and political surplus value for the study of borders and migration. We do so by studying processes of migrantisation in three contexts: (1) the Schengen visa regime, (2) policies

aiming at the 'integration' of migrants and (3) the government of mobility at Europe's southern frontier. Each case allows us to highlight particular aspects and analytical advantages of our definition of a migrant. They also illustrate how our definition can be operationalized. In brief, the first question to be raised in any research on 'migration' is who is (not) enacted as a migrant in the situation under study and how and through what kind of practices of border and boundary-making is this migrantisation done?

To answer this question, scholars should identify and study those instances in which either human mobility or the presence of some people are problematized and targeted as 'migration' in one way or another. These instances may be found in sites of border and mobility control, as illustrated by our first and third case. In such cases, scholars should focus their investigation on the 'embodied encounters' between mobile subjects and actors charged with controlling their mobility as it is in these encounters at border check posts, migration administrations, consulates etc. that (some) people are enacted as migrants through routinized bureaucratic assessments, administrative practices and related dialogues of action (Scheel, 2019: 96-102).Instances of migrantization that are primarily animated by processes of boundary-making feature, in turn, particular epistemic registers and related practices of knowledge production, as highlighted by our second case. The crucial analytical task is then to study the processes of migrantisation at work in these situations, the discourses, categorizations, taxonomies and knowledge regimes they rely on, the processes of racialization they feature, their complex relationships to with class, age, gender, sexual orientation, their implications for those labelled and targeted as migrants and how the latter may try to negotiate, escape, defy or openly resist their migrantisation. By attending to these aspects, scholars will be able to show that processes of migrantisation are not only heterogeneous and contingent, but also relational and contested.

## 4.1. The Schengen visa regime: enacting migrants, part I

Our first example concerns the EU's visa regime. Visa policies are one of the oldest techniques to outsource border controls beyond national demarcation lines. The imposition of a visa requirement enables the pre-screening and pre-selection of travelers before their departure (Zampagni, 2016). In the context of the Schengen visa regime, the criteria for the imposition of a visa requirement on a specific country evaluate its population in terms of risks, 'relating inter alia to illegal immigration, public policy and security' (Council, 2001: 3).[6] As the global map of the Schengen visa regime shows, this partition of the world in 'risky' and 'trustworthy' populations reflects geo-political asymmetries and socio-economic inequalities. However, not only high-income countries are exempt from a visa requirement, but also former white settler colonies in Latin America i.e. countries with a large share of the population of European origin. This seems to suggest that the imposition of a visa requirement is also informed by racializing discourses just as it is fraud with colonial histories.

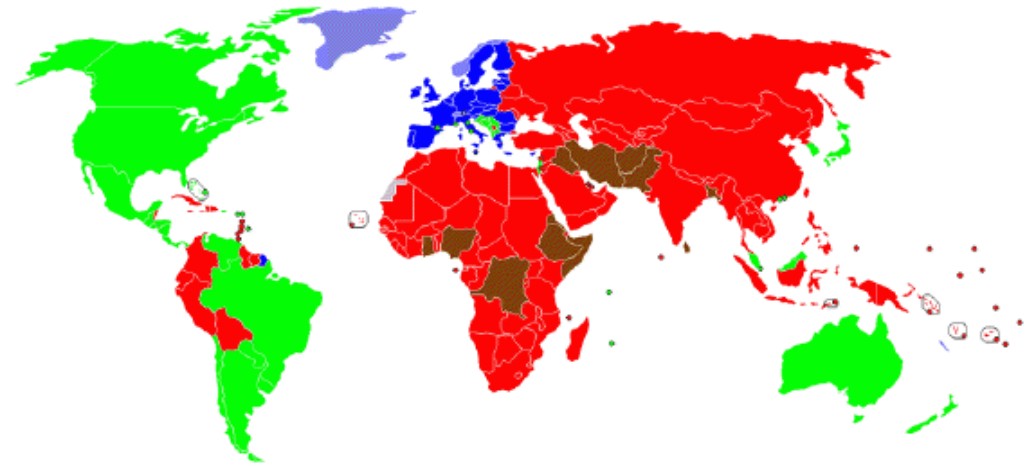

Official map of the Schengen visa regime: citizens of countries coloured in red need a visa to enter the Schengen area (coloured in blue) for a period of up to 90 days. Citizens from countries coloured in green are exempt from a visa requirement. Source: https://ec.europa.eu/home-affairs/policies/schengen-borders-and-visa/visa-policy_en (25.01.2022).

For all those who are subject to a visa requirement, the border is first enforced in the consulates in their country of residence - and thus long *before* they have reached the EU's geopolitical borders. In the application procedure, the presumption of innocence is reversed: It is the applicant who has to prove in an interview and through the provision of numerous documents that – contrary to the statistical knowledge which justified the imposition of a visa requirement in the first place – she does not pose a migration or security threat (Bigo & Guild, 2005: 250). Hence, visa applicants are subjected to a culture of institutionalized distrust when they apply for a Schengen at the consulates (Scheel, 2017). In practice any visa applicant will be denied access to mobility if she cannot convince consular staff of her 'will to return' to her country of departure.

If the visa application of a young man seeking to visit his brother in Europe is rejected he will receive a standardized letter stating that his intention to 'leave the territory of the Member States before the expiry of the visa applied for' could not 'be ascertained' (EP and Council, 2009: 12; emphasis added). In the moment his visa application is rejected the young man is enacted as a migrant by consular staff though he has never crossed a geopolitical border. Consular staff's practices like posing questions about a person's purpose of stay or verifying the authenticity of documents are performative because they bring into being and perform the very subject they seek to govern: a 'migrant'. The example of the consulate thus highlights the temporal and imaginary aspects of processes of migrantisation. People like the young man wishing to visit his brother living in Europe are denied a Schengen visa because they are suspected of becoming a migrant. In the eyes of consular staff, they are embodying a migration risk. Through this anticipatory risk assessment millions of people are enacted as potential migrants that have to be immobilized and kept in place. Importantly, this instance of migrantization is not captured by nation-state centered definitions of migration that posit the crossing of international borders as the central definitional criteria of a migrant. This central feature of statist conceptions of migration plunges the former in a deep epistemic crisis in the moment that practices of border control 'are no longer entirely situated at the outer limits of territories, [...but] dispersed a little everywhere' (Balibar, 2003: 1). In the consulates it is not the actions of an individual by which a person makes herself unilaterally a migrant, as assumed by state-centric definitions of a migration as movement from one national container to another one. Rather, what counts are the bordering practices of countless street-level bureaucrats charged with controlling human mobility that enact (some) people as migrants. It is thus, ironically, the de-localization of border controls beyond the edges of nation-states that brings to the fore the methodological nationalism of statist definitions of a migrant by plunging them into an epistemic crisis.

In contrast, our alternative definition of a migrant replaces the crossing of international borders with a focus on the border struggles that people who are treated as migrants have to engage in to move to or stay in a desired place. In the case of the Schengen visa regime, border struggles are implicated by an unpredictable regime of institutionalized distrust that renders mobility to Europe as a scarce resource through the introduction of an entry-ticket (a Schengen visa) whose receipt is subject to requirements that do not correspond to the living and working conditions of a large share of the local population (Scheel, 2017). Hence, many people engage in various tactics and practices in order to appropriate a Schengen visa within and against this vast control apparatus. They may for instance provide manipulated documents like job contracts or bank statements that support fictive biographies of people considered as 'bona fide' travelers by consular staff (Scheel, 2019). In these border struggles people try to appropriate an entry-ticket to Europe by escaping their migrantisation by the Schengen visa regime.

## 4.2. The integration paradigm: enacting migrants, part II

The problematization and government of immigrants' children and grandchildren as 'second' or 'third generation' migrants offers another illustration of the imaginary dimension of processes of migrantisation. People labelled as such have never left their 'country of usual residence' and do therefore not qualify as migrants according to the UN-definition (Schinkel, 2013). However, since the emergence of the integration paradigm as a central cornerstone of migration policies the descendants of immigrants are addressed as 'second' and 'third generation' migrants by 'integration policies', even if they hold the citizenship of their country of residence (Guild, 2009: 12-13).

What is problematized in case of the integration paradigm is not so much the mobility of people labelled as migrants but their presence. This shows that the 'sending-off to an elsewhere' accomplished by processes of migrantisation does not always revolve around the crossing of national borders (Tudor, 2018: 1064). Through discourses and practices that treat them as if they have just arrived, people labelled as 'foreign born' (in the UK), 'person with migration background' (in Germany) or 'second' or 'third generation' migrants, people labelled as such are held in a perpetual state of arrival (Boersma & Schinkel, 2018). They are subjected to a life-long apprenticeship they have to serve in order to become full, legitimate members of an imagined (national) community of shared values (Bridget Anderson, 2013). By tying citizenship to a racialized politics of belonging the integration paradigm renders citizenship a 'virtue' (Schinkel, 2010) or 'a faculty to be learned (Bridget Anderson, 2013: 100).

This redefinition of citizenship as virtue has very real consequences for people labelled as migrants: they have to *earn* formal citizenship and permanent residency (Schinkel, 2010: 272). In practice, they have to fulfil ever longer lists of acculturation and to constantly prove their moral worthiness and loyalty to the imagined community of shared values. This meritocratic understanding of citizenship as something to be earned also becomes manifest in the introduction of 'citizenship tests' across Europe since the 1990s (de Leeuw & van Wichelen, 2012). However, any process of integration presupposes a process of differentiation. In the context of Europe, this prior differentiation rests on a distinction between a (national) community of shared values and a culturally different, socially deficient subject in need of 'integration' (Schinkel, 2013). What the integration paradigm illustrates is that the labelling of people as ('second' or 'third generation') migrants constitutes a practice of boundary-making that is accomplished through the allocation of individuals to categories of difference (Grommé & Scheel, 2021) . The term 'migrant' – including its countless extensions and variations from 'third-generation of foreign-origin population' (Grommé & Scheel, 2021) to 'person with migration-background' (Renard, 2018) – operates as a performative category through which either the mobility or the presence of people labelled as such is enacted as a problem of government requiring close monitoring and interventions of regulation and control.

Again, it is informative to consider who is not enacted as a migrant in this context. The fact that not all newcomers or their children are problematized as 'migrants' in need of 'integration' indicates that the label 'migrant' refers to a racialized subject. Whereas new arrivals from Australia or the United States and their offspring are usually not considered as migrants in need of 'integration' in Europe, Algerian immigrants in France or Turkish 'guest workers' in Germany and their offspring are persistently labelled as ('second and third generation') migrants (Guild, 2009: 12). The salience of public debates on 'forced marriages', 'genital mutilation', 'honour killings', or the banning of 'burqas' and in some cases even the wearing of headscarves points to the importance of gender and family norms in anti-Muslim  racisms informing practices of boundary-making that enact Muslims living in Europe as eternal migrants, that is as socially deviant, deficient subjects who are in need of integration in an imagined community of shared values revolving around gender equality, sexual tolerance and laicism (Bonjour & Kraler, 2015; de Leeuw & van Wichelen, 2012; Fassin, 2010; Korteweg, 2017; Razack, 2004; Schinkel, 2013; Yilmaz, 2015). However, people who are labelled and addressed – in one way or another – as second, third etc. generation 'migrants' in need of integration often refuse, openly reject and subvert and invert these labels (Grommé & Scheel, 2021; Wimmer, 2013). These are precisely the kind of 'border struggles' in regards to practices of boundary-making that we want to highlight with our definition of a 'migrant'.

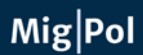

To analyse processes of migrantisation implicated by the integration paradigm scholars should therefore play close attention to how difference is produced and translated into 'otherness' (Meissner & Heil, 2020), particularly in practices of knowledge production like statistics (Grommé & Scheel, 2021; Renard, 2018; Schinkel, 2013). They should also attend to how these processes of othering translate into legal norms, practices of bordering and 'integration policies' and how these, in turn, affect and discriminate against people targeted as migrant 'others' in need of integration. Hence, 'any claim and practice that concerns integration should be [made] the object of research, rather than [accepted as] the project of research', as Leila Hadj Abdou (2019: 1) rightly summarizes this analytical stance. Finally, these analyses should consider any tactics and practices through which the labelled might contest or defy being labelled as a 'migrant', for example through the strategy of 'normative inversion' (Wimmer, 2013: 123), that is, the appropriation of their alleged otherness as something positive What these analyses will discover is that processes of migrantisation implicated by practices of boundary-making do, in most cases, not operate along a simple binary distinction between 'native' citizens and migrant 'others'. Rather, migrantisation is often a matter of degrees, as related practices and processes of bordering and boundary-making mobilise complex and shifting taxonomies, indexes, categories and classification systems.

## 4.3. Bordering the European space of mobility: enacting migrants, part III

The workings of different migration categories, and their varying impact on people's lives and journeys, highlight another important dimension of processes of migrantisation, namely their contingency and situatedness. Migrants are subjected to multifarious mechanisms of bordering and containment along their routes and these mechanisms interrelate with migrants' changing juridical status. To show this, we follow the trajectory of non-European 'migrant workers' escaping the war in Libya and contrast their geographically varying enactment as migrants with the geographically equally varying treatment of migrant workers escaping the economic crisis in Southern Europe.

In 2011, when attempts to overthrow the Gaddafi regime developed into a full-fledged war, almost one million people crossed the border to Tunisia. Most Libyans were hosted by Tunisians through so-called 'popular chains'. The thousands of 'migrant workers' from various Sub-Saharan countries who had been living in Libya were, in contrast, exempt from this hospitality. Most of them spent several months in Choucha refugee camp which was opened by UNHCR, holding up to 22.000 people in peak-times (Tazzioli, 2015: 102-114). In Choucha, UNHCR examined the asylum claims of these war escapees. Since they had not fled a war in their country of origin, most applicants were rejected and considered as 'people not of our concern' by UNHCR.

Faced with the choice to stay in Tunisia under precarious conditions of illegality or to return to their often war-torn, crisis-ridden countries of origin many non-Libyan war escapees decided to move on to Europe, crossing the Mediterranean in overcrowded boats. The humanitarian border spectacle in the Mediterranean occludes the systematic stranding and illegalisation of the rescued once they have reached Europe. Due to the Dublin III Regulation, the rescued war escapees had been subjected to the spatial restriction of applying for asylum in the Schengen member state through which they had entered Europe, thus being chased around Europe as illegalised asylum seekers (Picozza, 2017).

What the trajectory of non-Libyan war escapees illustrates is how migrants, during their journeys, are subjected to different bordering processes that enact and govern them accordingly – as economic migrants, as rejected refugees, as bodies to be rescued, as irregular secondary movers etc. Hence, the trajectory of people from Choucha shows how administrative practices enact the same person as a migrant in temporally and geographically varying ways, depending on the spaces of governmentality the person traverses (Tazzioli, 2015).

Yet, statist conceptions of migration overshadow, first, how particular bordering practices enact migrants in spatially and temporally varying ways and, secondly, that the enactment of some people as migrants always occurs in relation to others whose mobility and presence are normalized. To illustrate this relational aspect of processes of migrantisation we contrast the treatment of 'migrant

workers' escaping the war in Libya with the treatment of European citizens escaping economic crisis in Southern Europe.

Since the economic crisis started in 2008, tens of thousands of mostly young people have left Southern Europe to look for jobs and better living conditions elsewhere. While the majority has moved to Northern Europe, some have escaped the economic crisis by moving to African countries, most notably from Spain to Morocco, but also from Italy and France to Tunisia or from Portugal to Angola.[7] Many accept to work in deskilled jobs, for instance in call centers in Tangier and Rabat.[8] Most of these young 'migrant workers' enter Morocco and Tunisia as 'tourists' and live and work there as 'overstayers' beyond the period of three months they are allowed to stay without a visa.[9] While this praxis qualifies them as 'illegals', these young Europeans are usually not even considered as 'migrants', nor do they identify as such. They consider themselves 'expats' – a term which is exclusively used for 'European or North American nationals who move abroad, mostly for work-related reasons, including the former colonies' (Fechter & Walsh, 2010: 1199). Hence, the notion of the 'expat' emerges as a device of conceptual bordering assuring that neither the mobility nor the presence of white Westerners is problematized as 'migration' (this point has for instance been confirmed by: Kunz, 2020).

This distinction along lines of race and origin is reflected by the differential treatment both groups receive in the same space of governmentality: whereas new arrivals from Spain rarely encounter any problem to settle and work in Morocco people from Sub-Saharan countries face regular police controls, raids and deportation across the Algerian border (Human Rights Watch, 2014). Likewise, 'expats' from Italy and France in Tunisia are rarely asked for papers by the police, landlords or employers, while war escapees from Libya are regularly arrested. This differential treatment highlights that the problematization of certain individuals as 'migrants' operates not only in spatially and temporally contingent ways, but also in relation to others, whose mobility and presence are constituted as 'unproblematic'. To undo statist understandings of migration, it is therefore key to (1) interrogate who is problematized, othered and racialized as a migrant *here* and *now* and (2) to account for the fact that people are enacted as migrants (1) in geographically varying and temporally contingent ways.

## 5. Conclusion

One of the main epistemic and political stakes which underpin the question "who is a migrant?" consists in not seeing migration like a state. In this article we have problematized and challenged statist understandings of migration that, by adopting the nation-state point of view of spatial mobility, conceive of migrants as ready-available objects of research. To counter both the methodological nationalism and the ontologization of migrants implicated by statist understandings of migration we have developed an alternative conception of migration that highlights the constitutive role that nation-states' bordering practices play in the enactment of *some* people as migrants. To conclude, we clarify three aspects of this conception of migration to dispel potential criticisms that may be directed against it.

First, some people may object that our definition, due its focus on the vague notion of 'struggle', is rather imprecise and difficult to operationalize as it gives way to all sorts of ambiguous and marginal cases. While we do not deny that such cases exist, we would like to reply with the following two points. First, also the most prominent existing definitions of migration, such as the state-centered UN-definition of a migrant 'as a person who moves to a country other than his or her usual residence for a period of at least a year' (UN, 2002: 11) are haunted by limit cases. Although they would qualify as migrants according to this definition, official guidelines recommend to not include cross-border workers, diplomatic and military personnel and their dependents and nomads into official migration statistics (UNSD, 1998: 13). Furthermore, it has been shown that the use of data on country of birth or citizenship for the production of migration statistics implicates the migrantization of millions of people who have never crossed an international border but whose country of usual residence has changed due to geopolitical repercussions such as the dissolution of Yugoslavia or the Soviet Union (Gorodzeisky & Leykin, 2021). However, we believe – and this is our second point –that such liminal cases allow scholars – precisely because of their liminality, ambiguity and contingency – to highlight the implications of the 'national order of things' for people who are labelled and targeted as migrants

in order to show that the real problem is not migration, but the volatile and violent nation-state system generating this phenomenon in the first place.

Secondly, by de-naturalizing the national order of thing and drawing attention to the making of migration, our definition unsettles the migrant/citizen divide. Indeed, the racialization of some people as "migrants" has historically been consolidated in opposition to the "citizen": the question "who is a migrant?" can be answered only by interrogating who is enacted and racialized as a migrant here and now. Thus, by destabilizing and fracturing the notion of the "migrant", our definition also indirectly unsettles the category of "citizen".

Finally, by placing the bordering practices of nation-states at the center of attention we do not intend to overlook other factors that contribute to the production and government of migration – such as processes of migrant labor exploitation or geopolitical power asymmetries. Rather, our point is to stress that these factors are often mediated and articulated by bordering practices of nation-states and what we have called, more broadly, 'the national order of things' (Malkki, 1995). This implies, however, that the bordering practices of nation-states cannot be taken as isolated self-standing objects of critique. On the contrary, when studying processes of migrantization we should also explore how the national order of things is situated within a geopolitical context that is characterized by profound asymmetries in terms of access to mobility and how class, gender, (dis-)ability, sexuality, 'race' and nationality articulate each other in determining restrictions to freedom of movement. In this way, scholarly engagement with migrant struggles and processes of migrantization can provide an analytical angle for studying current transformations in regimes of government and capital accumulation. Learning not to "see like a state" (Scott, 1999) is ultimately the primary endeavor that the analytical lens of migration invites to engage in.

## 6. Endnotes

You must include endnotes. Author note: endnotes are below due to formatting settings of Microsoft Word

## Acknowledgements

Acknowledgements should follow immediately after the conclusion. Author note: will be included after article acceptance

**Funding information.** Authors are required to provide funding information, including relevant agencies and grant numbers with linked author's initials. Correctly-provided data will be linked to funders listed in the Fundref registry. To be added after peer review

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

# Endnotes

[1] There exist of course other definitions of a migrant. These are, however, not less state-centric than the UN-definition. For instance, the 'IOM defines a migrant as any person who is moving or has moved across an international border or within a State away from his/her habitual place of residence, regardless of (1) the person's legal status; (2) whether the movement is voluntary or involuntary; (3) what the causes for the movement are; or (4) what the length of the stay is' (cited from: https://www.iom.int/key-migration-terms#Migrant on 18th November 2016).

[2] See: https://www.merriam-webster.com/dictionary/immigrate (25.06.2021).

[3] In this article we refer to both immigrants and emigrants when we say 'migrants'. The reason is that emigration and immigration are the inseparable two sides of the same coin as Abdelmalek Sayad (2004) has emphasized time and again. A migrant is thus both an immigrant and an emigrant at the same time. To separate the processes of immigration and emigration analytically, means adopting either the perspective of the country of origin or of the receiving country. Such a statist conception of migration is, however, precisely what we criticize and abandon in this article.

[4] In the following we use the idiom of 'enactment' to emphasize that 'migrants' do not exist as given realities. They have to be brought into being and performed through a range of re-iterative practices that constitute some people as migrants by addressing them as such. Put differently, practices of bordering and knowledge production enact (that is: bring into being and perform) that to which they refer. Such understanding of enactment as an alternative term for performativity has been developed in Science and Technology Studies (Mol, 2002; Scheel, Ruppert, & Ustek-Spilda, 2019). In the following we loosely refer to this notion of enactment to highlight the contingent and contested character of what we call processes migrantisation.

[5] While some authors use borders and boundaries interchangeably, we distinguish between the two term to emphasize that they refer to related, but ultimately different aspects of processes of migrantisation. Bordering practices are practices of statecraft that aim at the constitution and preservation of the nation-state as a territory comprising a political authority and a bounded group of people, that is, the national citizenry enjoying the exclusive right to reside, live and work on this territory. Processes of boundary-making operate, in contrast, more on the discursive and symbolic level, albeit with real-world effects. As Andreas Wimmer (2013) and others highlight, processes of boundary-making constitute differences between groups along lines of ethnicity and play a key role in the constitution of imagined communities on the national level as well as related politics of belonging (cf. Brubaker, 2009; Wimmer, 2013; Yuval-Davis, 2008 [1997]).

[6] The member states of the Schengen area maintain a common visa regime for short term visa with a validity of up to 90 days. Since there are no border controls between the 26 member states of the Schengen area, a Schengen visa usually allows its holder to travel across all member states of the Schengen area. People have to apply for a Schengen visa at the consulate representing the member state in which they which to spend most of the time of their stay. While the EU has tried to 'harmonize' the rules and procedures for application and decision-making procedures through s shared Visa Code (EP and Council, 2009) as well as various handbooks, both procedures are still characterized by a vast heterogeneity and inconsistency across the approximately 3.500 consular posts that the 26 Schengen member states maintain worldwide (cf. Infantino, 2016; Scheel, 2019).

[7] See: http://diasporaenligne.net/immigration-le-maroc-accueille-des-travailleurs-pauvres-espagnols/ (01.08.2021)

[8] See: https://www.jeuneafrique.com/49710/politique/maroc-espagnols-cherchent-travail/; http://heindehaas.blogspot.de/2012/07/europeans-looking-for-greener-pastures.html (04.08.2021).

[9] Most of the 20 Spaniards interviewed in Tangier by Lotte Rooijendijk (2013) for her Master thesis reported for instance that they work and reside in Morocco as 'tourists', though this status neither entitles them to employment nor to stay for longer than 90 days, a period most interviewees had exceeded at the time of the interview. On this point see also: http://lejournaldusiecle.com/2013/06/12/quand-les-espagnols-entrent-clandestinement-au-maroc-pour-y-travailler/ (10.10.2021).