# Peer review of "Who is a Migrant? Abandoning the Nation-state Point of View in the Study of Migration"

_Migration Politics_

## Round 1 · Referee Report · Bridget Bridget Anderson (Referee 1) · 2021-10-1

Strengths

  1. Conceptually advances autonomy of migration approach;
  2. Combines insights from different literatures;
  3. Makes a novel proposal for definition of 'migrant';
  4. Innovative response to the problem of methodological nationalism.

Weaknesses

  1. Lack of explanation of why 'groupism' constitutes methodological nationalism;
  2. Assumption of a singular 'migrants' perspective'.
  3. Borders and boundary making, and, relatedly, the 'nation' element of nation state insufficiently explored.

Report

The journal's acceptance criteria are met. This is potentially an important conceptual intervention. It is an ambitious and thoughtful article that attempts to move away from a nation state understanding of migration and towards a ‘migrant’ perspective. In so doing it takes a broader understanding of who is ‘migrantized’ i.e. made into a migrant, including people who find themselves unable to move and centralizing struggles: ‘a migrant as a person who, in order to move or to stay in a desired place, has to struggle against bordering practices that are implicated by the national order of things’.
This step is potentially a really important contribution, but the authors have not anticipated some of the obvious criticisms that could be made of this definition.
1. There is not a singular ‘migrants’ perspective’ – at the very least this should be recognised by using the plural. What does this multiplicity of perspectives entail – particularly given that people on the move have all sorts of reasons for having heightened national subjectivities;
2. IF struggle is centralised, then presumably not all those who are subject to immigration controls are migrants (unless having to apply for a visa counts as struggle). This is potentially an interesting analytic move that merits noting even if there is not the space to discuss it.
3. The terms ‘bordering’ and ‘boundary making’ are insufficiently distinguished from each other. I agree, boundary making is important, particularly with reference to race, but this is implicit rather than engaged with. If the definition of ‘migrant’ is to include racialized so-called ‘second generation’ who are citizens then boundary making becomes more important – and perhaps should be included in the definition (though mightn’t racialised citizens resist being labelled as ‘migrants’ however it is defined?).
Why is ‘groupism’ methodological nationalism? Surely it doesn’t have to be – one can be regarded as part of the Jewish diaspora without associating oneself with the state of Israel. This needs more explication.
Section 2 assigns the conflation of migration with international migration to history – but this isn’t actually the case (eds Ramirez et al. 2021 Precarity and Belonging: Labor Migration and Non-Citizenship) and there are clear comparisons to be made between the reluctance of British officials to designate intra Imperial mobility ‘migration’ and the European Commission’s vocabulary of ‘mobile citizens’.

Requested changes

  1. Reflect on the suggestions re the definition and anticipate some of the criticisms that might be made;
  2. Be clearer about who might, and who might not, count as a migrant under this definition;
  3. Clarify the distinction between bordering and boundary making;
  4. Give some thought to the consequences of multiplicity of perspectives, including those of people who might resist being migrantized;
  5. Give an explanation of why groupism necessarily constitutes methodological nationalism

---

## Round 1 · Referee Report · Ayse Dursun (Referee 2) · 2021-10-12

Strengths

  1. Seeking to overcome the state-centered approach to "the migrant" and migration by shifting the focus to migrants' struggles to move to or stay in a place
  2. While doing so, it does not lose sight of political institutions and norms, most importantly the nation state
  3. The selected empirical cases are well-fitting and support the authors' arguments/theses

Weaknesses

  1. The authors need to revise their introduction which can be shortened and condensed so that their own position (e.g., their own definition of a migrant and that they in fact do not suggest to think away the state from migration studies) becomes clear much earlier in the article (in parts, it became clear to me only on page 7-8 -9)
  2. The article should be proofread (language check, harmonization of citation style)
  3. There is generally potential for streamlining the article to represent the theoretical considerations in a more neat and tidy fashion and avoid repetitions

Report

Generally, I believe that article entails valuable considerations which would make an important contribution to the field. I would recommend this article for publication after some revisions have been made; please find the requested changes below.

Requested changes

P. 1 - - > (Bridget Anderson, 2017: 1532) - - > remove „Bridget“ from the citation
P. 2 - - > „a tool for inquiry’ (1535)“ - - > (ibid., 1535)? Depending on the style of citation
P. 2 - - > „expose and challenge the ‘the nation-state point of view“  remove the second „the“ (in bold)
P 2 - - > „We engage with this question to expose and challenge the ‘the nation-state point of view of spatial mobility’(Favell, 2007: 271) which underpins the framing of migration as a problem requiring constant monitoring as well as governmental interventions of regulation and control.“ - - > This is also the EU point of view judging by restrictive migration policies on the supranational level of governance; could be briefly mentioned.
P. 2 - - > “because they are rest“ - - > remove „are“
P. 2 - - > „suggestions of how“ - - > suggestions on how (?) The language of the article is very good, but should still be proofread
P. 3 - - > “they do have very real consequences“ - - > they have real consequences (?) (see my suggestion above re. proofreading)
P. 3 - - > “analytical and political surplus value“ - - > contribution? or added value?
My impression is that the introduction could be much shorter; especially the paragraph starting with “The three examples we chose relate…“ (p. 3) go into detail about the three empirical examples. I think it would suffice if the authors briefly introduced/mentioned the three empirical examples they wish to elaborate on but save the discussion/findings for later on.
P. 4  „is that it did not did not distinguish“ --> delete „did not“
Citation style must be harmonized  e.g., (Chernilo, 2011: 99); (Brubaker, 2002, p. 164)
P. 5 - - > „statist conception of migration“ - - > does it mean state-centered or imposed by the state?
I agree with the authors on their observations on ontologization of “the migrant”, naturalization of the nation-state and securitization which results from the first two; however, this does not change the fact that nation-state play a crucial role in constructing migrants as migrants through their border and migration policies - - >  leaving the nation-state out of the equation would relativize this fact - - > for example if we get rid of the “container-thinking” in research, how can we assess the nation state’s role in constructing certain subjects as “migrants” and “third-country nationals” and others as “nationals” or “EU citizens”? - - >  *** This is a note added later, after reading the article up to p. 9; it becomes clear that the authors do not suggest to think away the state but to shift the focus to migrants’ struggles which are still embedded in the broader context of “national order of things” - - > I think this is a very valuable suggestion and should be made explicit earlier in the article
The concept of “security” is introduced on p. 5 which should be introduced earlier if it is a key concept for the current paper as it seems to be
P. 6 - - > “The critical security studies scholar Didier Bigo underlines this effect of methodological nationalism, arguing that ‘the securitization of the immigrant as a risk is based on our conception of the state as a body or a container of the polity’ (2002: 65).“ - - > But is there not a difference between: (1) nation states considering themselves a marker (or container) of a national territory, polity and population vs. social scientists saying that nations states consider themselves a marker (or container) of a national territory, polity and population - - > would the social scientist be reproducing the container logic when she notes that nation states function as containers? - - > To me it seems a more convenient way forward to suggest that we should problematize methodological nationalisms without downplaying the nation state’s historical and structural importance in constructing migrants; I think the authors are not really suggesting to think away the state (if I am not mistaken), this maybe made more clear (authors explain it convincingly on p. 7, paragraph 2 starting with ‘While we share Dahinden’s…“ or when they write on p. 8 “What distinguishes migration from other forms of mobility is that it is the fabrication of clashes with practices of statecraft“; they can make this point more explicit already earlier in the text as these assumptions seem to distinguish the authors from others and capture the quintessence of the article).
P. 6 - - > “There is Since the…“ - - > delete „There is“
P. 8 “stasis“ - - > states?
P. 8 „as a intelligible reality“ - - >  as an…
P. 9 „from the literatures discussed above“ - - > from the works?
P. 9 „we therefore understand a migrant as a person who, in order to move to or stay in a desired place, has to struggle against bordering practices that are implicated by the national order of things“ - - > I think this can be disclosed much earlier in the article (in the intorduction?) to facilitate a better understanding of their authors‘ undertaking
P. 9. „has to struggle against bordering practices that are implicated by the national order of things“ - - > against state-imposed bordering practices?
P. 9 “First,, our…“ - - > double commas
p. 10 „rely on, the processes“ - - > without comma
P. 11 “If the visa application of a young man seeking to visit his brother in Europe is rejected because his intention to ‘leave the territory of the Member States before the expiry of the visa applied for’ could not ‘be ascertained’ (EP and Council, 2009: 12; emphasis added).“ - - >  sentence incomplete and no emphasis added
P. 11 „it are“ - - >  I am not familiar with this use, the authors may be correct but should still check
Do the authors need „part I“, „part II“, „part III“? - - >  these are rather selected cases or examples rather than the parts of a whole (even if, the authors do not intend to exhaust all parts)
The three examples are carefully selected and the authors are very convincing in their elaborations of each case
P. 14 “narrow focus on international migration“ - - >  internal?

---

## Round 1 · Referee Report · Anonymous (Referee 3) · 2021-10-27

Strengths

  1. It puts forward a novel and interesting conceptualisation of a migrant.
  2. It engages well with critical migration studies and the AoM literature.
  3. It selects three very relevant empirical cases to highlight the practices of migrantisation in place.

Weaknesses

  1. It does not interact enough with more mainstream literature that addresses bordering and migrantisation practices (see more details in the report below).
  2. The definition of migrant could lead to a more fine-grained operationalization, distinguishing the multifaceted results of bordering practices (different types of migrants).
  3. The selected cases are relevant and their choice seems to be based on a most similar cases method. They are not, however, particularly puzzling. Are there cases in which contradicting practices emerge? What happens to the conceptualization if a most different cases method is used?

Report

I have read "Who is a Migrant? Abandoning the Nation-state Point of View in
the Study of Migration" with great interest. The topic is highly relevant for Migration Politics, the acceptance criteria are met and, if published, it will contribute to a high degree to debates on migration policies, practices and processes. The proposed conceptualisation of who is a migrant is novel and could lead to interesting follow-up work. I suggest minor revisions, hoping that the comments will be as constructive as possible and they will strengthen the paper even further. I will summarise my decision further below.

While a literature review cannot cover the totality of migration research, and while section 2 is very well written, the authors could acknowledge the fact that also some more mainstream research (literature on the external dimension of EU migration policy, debates on practices in migration studies, in EU studies and in IR research, as well as the decentring debate in IR and migration studies), focuses on borderwork and highlights a whole range of practices, rules and acts that “create the migrant” or, as the authors name it, “the making of migration”. Briefly interacting with this literature is particularly important if, as the authors argue, the paper aims at contributing both to the literature on Autonomy of Migration but also to mainstream literature. I agree that this literature is partly covered in the empirical section, yet a quick remainder could be included in section 2 as well.

The conceptualisation of "a migrant" is interesting and valuable. This, however, does not help distinguishing the multifaceted results of border practices which create many migrants, create hierarchies of deservedness and can be associated to the evolution of the definition of nation-state. Bordering practices produce, indeed, many types of migrants, not only one.

The selected cases are relevant and their choice seems to be based on a most similar cases method. They are not, however, particularly puzzling. Are there cases in which contradicting practices emerge? What happens to the conceptualization if a most different cases method is used? While the paper is theoretical, a quick reflection on this point could strengthen the validity and applicability of the conceptualization put forward in the contribution.

Finally, a number of typos should be corrected, although this will be done at a later stage during the production process.

---

## Round 2 · Author Response

To the reviewers and editorial team of MigPol,
First of all, we would like to thank you all for taking your time to review out manuscript and for the thoughtful comments and constructive suggestions we received. In the list of changes below we will briefly explain, how we have tried to address each point, in particular the points highlighted by the editor-in-charge, Leila Hadj Abou. Whenever possible, we have provided page numbers referring to the passages in which the changes have been implemented in the revised manuscript.
We did however not highlight the changes in a particular colour or the like as we could not find anything about this in the guidelines for authors, so we decided to submit a clean version of our revised manuscript.
We hope that we have addressed all comments to your satisfaction. Should you require any clarification or additional information please do not hesitate to contact us.
With best wishes,

Martina and Stephan

---

## Round 2 · List of Changes

List of implemented changes:

1. Reviewers have asked us to stress that there is not a singular ‘migrants’ perspective’ and that “at the very least this should be recognised by using the plural.”

Response: On page 2 in the introduction we have inserted a few lines which emphasise that there exist only a multiplicity of migrant perspectives because there are a myriad ways to be a migrant which are criss-crossed and shaped by lines of class, race, gender and so on. We have also tried to clarify that out definition of a migrant tries to emphasise this irreducible multiplicity with its focus on border struggles and its suggestion to begin the analysis by asking who is enacted as a migrant in this particular situation pertaining to the context under study. We have also added a short paragraph right after introducing our definition on page 9 of the revised manuscript to dispel this potential criticism.

2. Based on the reviews the editor notes, that if struggle is centralised in our alternative definition of a migrant, “then presumably not all those who are subject to immigration controls are migrants (unless having to apply for a visa counts as struggle).” While the editor acknowledges, that we will probably not have enough space to discuss all possible nuances and configurations, the editor invites us, nevertheless, to “anticipate some of the criticisms [and objections] that might be made” against our definition and to discuss some limit cases.

Response: We have added a few lines before and a short paragraph right after introducing our definition on page 9 in which we anticipate and try to dispel some of the doubts and criticisms that may be raised against our alternative definition of a migrant. In this new passage we (1) clarify what we mean with the notion of ‘migrant struggle’ in order to stress (2) that our definition does not intend to imply that all people subjected to border controls qualify as migrants according to our definition (3) because this requires that they are denied to move to or stay in a desired place because they are considered as ‘quintessential others’ vis-à-vis people considered as native/national citizens. We have also clarified in the section on the visa regime on page 11 that the young man is only enacted as a migrant in the moment that his visa application is rejected because he is considered as a ‘migration risk’. So not all people subjected to a visa requirement are migrantised, but those who are refused a visa.
In the new paragraph on page 9 we underscore, moreover, that our definition does not seek to reduce all migrants to one migrant condition. Quite to the contrary, it highlights the irreducible multiplicity of migrant conditions, perspectives and border struggles by inviting scholars to engage in a situated analysis and to begin their investigation by asking who is enacted as a migrant through what kind of practices of bordering and boundary-making in this particular situation I want to study? In this way our definition implicates a fracturing of the category of the migrant, thus resulting in an understanding of migration as something that is relational, contested and multiple.
Moreover, in the conclusion (pages 14-15) we point out that, while our definition may indeed open up space for difficult and ambiguous ‘marginal cases’ in regards to the question who should be considered as a migrant, (1) these marginal cases allow to invert the problem of migration and to scrutinize and problematize the implications of the national order of things for people who are migrantised. Furthermore, we emphasise (2) that the existing standard (state-centric) UN-definition of a migrant can only be maintained and operationalised by introducing all sorts of exceptions (for diplomatic staff, military personnel etc. that are not considered and counted as migrants) and methodological tricks.

3. To this regard, the reviewers and the editor have also asked us to “be clearer about who might, and who might not, count as a migrant under this definition and engage more with the fact that there is not a singular migrants perspective”.

Response: On page 9 as well as in the introduction we have inserted a few lines in which we explain and underline that our definition does not intend to reduce all migrants to one singular migrant condition. Quite to the contrary, it highlights the irreducible multiplicity of migrant conditions, perspectives and border struggles by inviting scholars to engage in a situated analysis and to begin their investigation by asking who is enacted as a migrant through what kind of practices of bordering and boundary-making in this particular situation I want to study? In this way our definition implicates a fracturing of the category of the migrant, thus resulting in an understanding of migration as something that is relational, contested and multiple.
On page 9 we also emphasise that not all people subjected to border controls or processes of boundary-making qualify as migrants. Only if they their right to reside in or move to a desired place is denied or called into question because they are considered and addressed as the ‘others’ of native/national citizens due to racialized notions of belonging and practices of bordering they qualify as migrants according to our definition. In the three examples that follow we have tried to clarify this further by discussing limit cases or ambiguous cases for each example, including examples in which ‘migrants’ contest being labelled and addressed as ‘migrants’ (see below).

4. They have also asked us to “add some lines on operationalization” i.e. to elaborate on how to operationalise our definition and to also “add some reflections if your ideas hold when taking some „harder“ cases.”

Response: We have updated and slightly extended the opening paragraph of the article’s third and last part on page 9-10 of the revised manuscript. In this paragraph we have provided some concrete instructions of how our definition can be put to use, namely by asking, as the first question of any research on migration, ‘who is (not) enacted as a migrant in the situation under study and how and through what kind of practices of border and boundary drawing is this migrantisation done?’ We have also provided a list of things and aspects scholars should look out for while engaging with this question. In addition, we have also added a few lines on how to operationalise our definition in particular contexts, for instance in relation to processes of boundary-making in context of ‘integration policies’, which require a different research strategy than conventional practices of bordering (page 13). Due to space constraints, we could only add a few reflections on ‘harder’ or more ambiguous cases in the conclusion (page 14-15).

5. We have been asked to “clarify the distinction between bordering and boundary making” and to also include boundary-making as a distinct element in our definition of a migrant.

Response: We have inserted a new endnote (5) on page 2 in the introduction, in which we explain that we distinguish between practices of bordering, which we understand as practices of statecraft enacting the nation-state as a bounded territory, jurisdiction and people, and processes of boundary-making which operate, in contrast, more on the discursive and symbolic level, playing a key role in the constitution of groups along lines of ethnicity and nationality. We have also added ‘processes of boundary-making’ as another element of our alternative definition of a migrant.
Building on these clarifications, we now highlight in section on integration that second, third etc. generation migrants refuse, contest and resist being labelled as a migrant or a ‘person with migration-background’ (page 12). We argue that these contestations and refusals are precisely one form of the border struggles we have in mind in relation to processes of boundary-making, highlighting the contingent and contested character of processes of migrantisation.

6. One reviewer rightly noted that we should provide “an explanation of why groupism necessarily constitutes methodological nationalism”

Response: We have addressed this comment on page 5 of the revised manuscript where we have added two lines. There, we explain that this form of methodological nationalism has become a cornerstone of ‘integration studies’, a whole branch of the field of migration studies, because [quote from revised manuscript] “groupism supports territorialised understandings of culture and (national) identity and the related conception of migrants (and their descendants) as ‘people not from here’ in need of integration.” We have also added references to two studies supporting this argument (Dumitru 2014 and Renard 2018).

7. The editor has asked us, in line some reviewers, to engage more with and acknowledge important “mainstream literature” that has criticised processes of migrantization.

Response: on page 8 of the revised manuscript we have inserted a new, short paragraph that refers to some important contributions, such as Roger Zetter’s labelling approach and his related critique of the fragmentation of the label of the refugee through the invention of new categories or the more general critique of the distinction between economic migrants and refugees/ forced migration as well as critiques of the dominant state-perspective in taxonomies of and distinctions between temporary vs. permanent, internal vs. international migration.

8. The editor-in-charge has asked us to “shorten the introduction and streamline/cut repetitions”

Response: We have tried to trim and streamline the introduction and to cut out any repetition throughout the manuscript, also to address the points raised above while keeping the manuscript close to the word limit of 12.000 words.

9. Finally, it was requested that we “harmonize the citation style” and do a final round of thorough proofreading.

Response: After liaising with the three editorial fellows we changed the citation style to APA 6th and updated the list of references. We also did a thorough proofreading. In particular we corrected all the mistakes and errors discovered by Ayse Dursun (report 2) whom we would like to thank for the close reading.

---

## Round 3 · Author Response

Dear Dr. Hadj Abdou,
Thank you very much for your careful reading of our resubmission and for recommending our article for publication. We are really happy about this outcome and that you were mostly satisfied with how we responded to the comments and suggestions of the three reviewers.

In regards to your final requests we would like to respond as follows:
1. in the abstract: “after framing of migration as problem of government, I would add: in need of intervention and control.” We tried to implement this but unfortunately due to lack of space (limit of 200 words for abstracts) were unable to do so.
2. Minor grammatical errors: we corrected these, thank you very much for the careful reading!
3. Request to highlight that also some mainstream migration researchers have at times pointed ‘towards the fact that there would be nothing such as migration without national boders’. Hence, you invite to consider and reference for example the works of Ari Zolberg and Andrew Geddes. Thanks for this observation. However, we do think that we have pointed this out repeatedly. For example, we cite Zolberg (1989) in the following passage on p. 8: “What distinguishes migration from other forms of mobility is that it is the fabrication of clashes with practices of statecraft. ‘It is precisely the control which states exercise over borders that de-fines international migration as a distinct social process’ (Zolberg, 1989: 405).” We also cite and reference other more mainstream scholars such Adrian Favell or Hein de Haas. However, to do justice to your point we have included a new reference to Christian Joppke’s (1998) work by citing him on page 5 in a new sentence included in the section on the three epistemological traps implicated by state-centric definitions of migration: “By representing nation-states as passive spatial units that are crisscrossed by migratory move-ments state-centred definitions of migration ‘obscures that the modern state and system of states have helped [and still help] to produce what they seek to contain: international migration’ (Joppke, 1998: 5).”
4. Your own critical stance on the integration paradigm: To be honest, we were not aware of your contribution to the debate triggered by Willem Schinkel’s book about integration research as a neo-colonial discourse and practice. We cited a passage from your article (Hadj Abdou 2019) published in Comparative Migration Studies to highlight that your proposition in regards to the migrant integration paradigm and its othering effects resonates significantly with our own agenda of making processes of migrantization the main focus of research instead of assuming ‘migrants’ as ready available objects of (integration) research.
5. Deletion of footnotes: We re-read all the footnotes, in particular those mentioned by you as potentially superfluous and dispensable (footnotes 1, 3 and 6). However, with the exception of footnote 6 on borders also being constitutive of ‘internal migration’ (and indispensable for its measurement), we believe that the other two footnotes provide important additional information that is crucial for fully grasping the scope of the argument we are trying to make.

We hope that have now addressed all comments to your satisfaction and would like to tank you once more for you constructive comments and repeated careful reading of our manuscript.

Kind regards,

Martina (Tazzioli) and Stephan (Scheel)

---

## Editorial Decision

unknown